# Targeted induction of a silent fungal gene cluster encoding the bacteria-specific germination inhibitor fumigermin

Maria Cristina Stroe[1,2], Tina Netzker[1†], Kirstin Scherlach[3], Thomas Krüger[1], Christian Hertweck[2,3], Vito Valiante[4], Axel A Brakhage[1,2]*

[1]Department of Molecular and Applied Microbiology, Leibniz Institute for Natural Product Research and Infection Biology (HKI), Jena, Germany; [2]Institute of Microbiology, Friedrich Schiller University Jena, Jena, Germany; [3]Department of Biomolecular Chemistry, HKI, Jena, Germany; [4]Leibniz Research Group – Biobricks of Microbial Natural Product Syntheses, HKI, Jena, Germany

**Abstract** Microorganisms produce numerous secondary metabolites (SMs) with various biological activities. Many of their encoding gene clusters are silent under standard laboratory conditions because for their activation they need the ecological context, such as the presence of other microorganisms. The true ecological function of most SMs remains obscure, but understanding of both the activation of silent gene clusters and the ecological function of the produced compounds is of importance to reveal functional interactions in microbiomes. Here, we report the identification of an as-yet uncharacterized silent gene cluster of the fungus *Aspergillus fumigatus*, which is activated by the bacterium *Streptomyces rapamycinicus* during the bacterial-fungal interaction. The resulting natural product is the novel fungal metabolite fumigermin, the biosynthesis of which requires the polyketide synthase FgnA. Fumigermin inhibits germination of spores of the inducing *S. rapamycinicus,* and thus helps the fungus to defend resources in the shared habitat against a bacterial competitor.

*For correspondence:
axel.brakhage@hki-jena.de

Present address: †Department of Biology, McMaster University, Hamilton, Canada

## Introduction

In their ecological niche, microorganisms interact with neighboring species in mutualistic or antagonistic ways. For many of their communication processes, they rely on chemical compounds (*Mithöfer and Boland, 2016*), often belonging to the group of secondary metabolites (SMs) (*Keller et al., 2005*; *Clardy et al., 2009*; *Brakhage, 2013*; *Netzker et al., 2015*; *Keller, 2019*). SMs show high chemical diversity, and their encoding genes mostly cluster together in a particular genomic locus (*Smith et al., 1990*; *Trail et al., 1995*).

The biosynthesis of SMs is regulated at various levels to ensure production of SMs in a relevant context (*Brakhage, 2013*). Several environmental signals have been identified as triggers of SM production, such as interactions of neighboring microorganisms (*Künzler, 2018*). In the artificial setting of a laboratory where natural triggers of SM production are missing, biosynthetic gene clusters are often not expressed and therefore their corresponding SMs remain undetectable (*Bergmann et al., 2007*; *Brakhage, 2013*; *Rutledge and Challis, 2015*). One approach that has proven to be particularly successful in stimulating microbial SM discovery, as well as avoiding redundancy often leading to re-isolation of known compounds, is mimicking the settings of a particular ecological niche through microbial co-culturing (*Netzker et al., 2018*). In particular, competitive inter-kingdom interactions have been exploited as a trigger for production of bioactive compounds (*Watanabe et al., 1982*; *Cueto et al., 2001*; *Moree et al., 2013*; *Khalil et al., 2019*). For example, several studies have reported the induction of compound biosynthesis in *Aspergillus* spp. by *Streptomyces*

(*Schroeckh et al., 2009*; *Wu et al., 2015*; *Yu et al., 2016*) or by *Mycobacterium* sp. (*Jomori et al., 2020*). Conversely, downregulation of fungal metabolite biosynthesis by *Streptomyces* spp. was also reported (*Verheecke et al., 2015*).

In our previous work, we discovered an inter-kingdom interaction between the soil-dwelling actinomycete *Streptomyces rapamycinicus* and the filamentous fungus *Aspergillus nidulans* (*Schroeckh et al., 2009*). We showed that the silent SM gene cluster encoding biosynthesis of orsellinic acid and its derivatives in *A. nidulans* is activated on physical contact of the fungus with the bacterium (*Schroeckh et al., 2009*; *Nützmann et al., 2011*; *Fischer et al., 2018*). We subsequently showed that the same bacterial strain also engaged in intimate interplay with the fungus *A. fumigatus*, in this case triggering expression of the silent polyketide gene cluster encoding the C-prenylated fumicyclines (*König et al., 2013*). In both instances, the SMs could only be isolated from a fungal-bacterial mixed culture, with no expression of the biosynthetic genes in axenic fungal cultures, which underlines the power of mixed cultures in awakening silenced SM production. In both cases, the ecological function of the produced compounds has remained elusive. However, it is important to unravel general principles of the interactions of microorganisms and to understand functional interactions of microorganisms in microbiomes.

Here, we now report an example where expression of biosynthesis genes of a novel fungal compound is triggered by a bacterium and where the compound then acts on the inducing bacterium.

## Results

### *S. rapamycinicus* triggers production of the polyketide fumigermin in *A. fumigatus*

Based on our previous discoveries that the biosynthesis of novel compounds can be triggered by co-culturing *A. nidulans* or *A. fumigatus* with *S. rapamycinicus* (*Schroeckh et al., 2009*; *König et al., 2013*), we analyzed the metabolic profile of the *A. fumigatus* isolate ATCC 46645 in co-culture with *S. rapamycinicus* for the presence of further metabolites. Using LC-MS analysis, we detected a new metabolite (1) in the mixed fermentation of *A. fumigatus* ATCC 46645 and *S. rapamycinicus,* but not after the addition of supernatant of an *S. rapamycinicus* culture to the fungal culture (*Figure 1A*, *Figure 1—figure supplement 6*). Other streptomycetes such as *S. iranensis*, *S. coelicolor* or *S. lividans* also induced the production of compound 1 during co-cultivation, but to a much lesser extent (*Figure 1A*). Comparative metabolic analyses by HPLC-HRESI-MS confirmed the presence of high amounts of compound 1 in the extract of the bacterial-fungal co-culture, and trace amounts in that of the axenic fungal culture. For compound 1 a molecular weight of *m/z* 194 amu was detected. The compound was isolated from the crude extract and its structure was elucidated by NMR analyses. A molecular formula of $C_{11}H_{14}O_3$ was determined by HRESI-MS. The $^{13}C$ and DEPT135 NMR data (*Figure 1—figure supplement 1B*, *Figure 1—figure supplement 2A*) indicated the presence of four methyl and seven double-bonded carbon atoms, six of which proved to be quaternary. The only methine proton (H-7) was shown to be in vicinity to one methyl function (H-8) by the respective H,H-COSY coupling (*Figure 1—figure supplement 2B*). HMBC couplings of the protons of C-2-CH$_3$ with C-1, C-2 and C-3, and of C-4-CH$_3$ with C-3, C-4 and C-5 revealed the presence of a pyrone ring. HMBC correlations between the C-6-CH$_3$ protons with C-5, C-6 and C-7 and between H-7 and C-5 disclosed the structure and the position of the side chain (*Figure 1—figure supplement 3B*).

Compound 1 represents an α-pyrone and was named fumigermin (*Figure 1B*, *Figure 1—figure supplement 4*). It is structurally similar to other microbial α-pyrones, including the fungal SMs nectriapyrone from *Piricularia oryzae* (*Motoyama et al., 2019*) and gibepyrone produced by *Fusarium fujikuroi* (*Janevska et al., 2016*). It is also similar to the bacterial germicidins (*Figure 2C*) originally isolated from *Streptomyces viridochromogens* (*Aoki et al., 2011*), but also produced by *Streptomyces* species *S. coelicolor* and *S. lividans* (*Song et al., 2006*; *Chemler et al., 2012*).

### Fumigermin biosynthesis requires the *fgnA* polyketide synthase gene

The induced production of a fungal metabolite prompted us to monitor the expression of potential biosynthesis genes in co-culture. The elemental composition of the metabolite revealed by its HRESI-MS-deduced molecular formula (195.1015 [M+H]$^+$, calcd. $C_{11}H_{15}O_3$ 195.1016) showed a lack of nitrogen atoms and therefore pointed towards a possible polyketide origin of the compound.

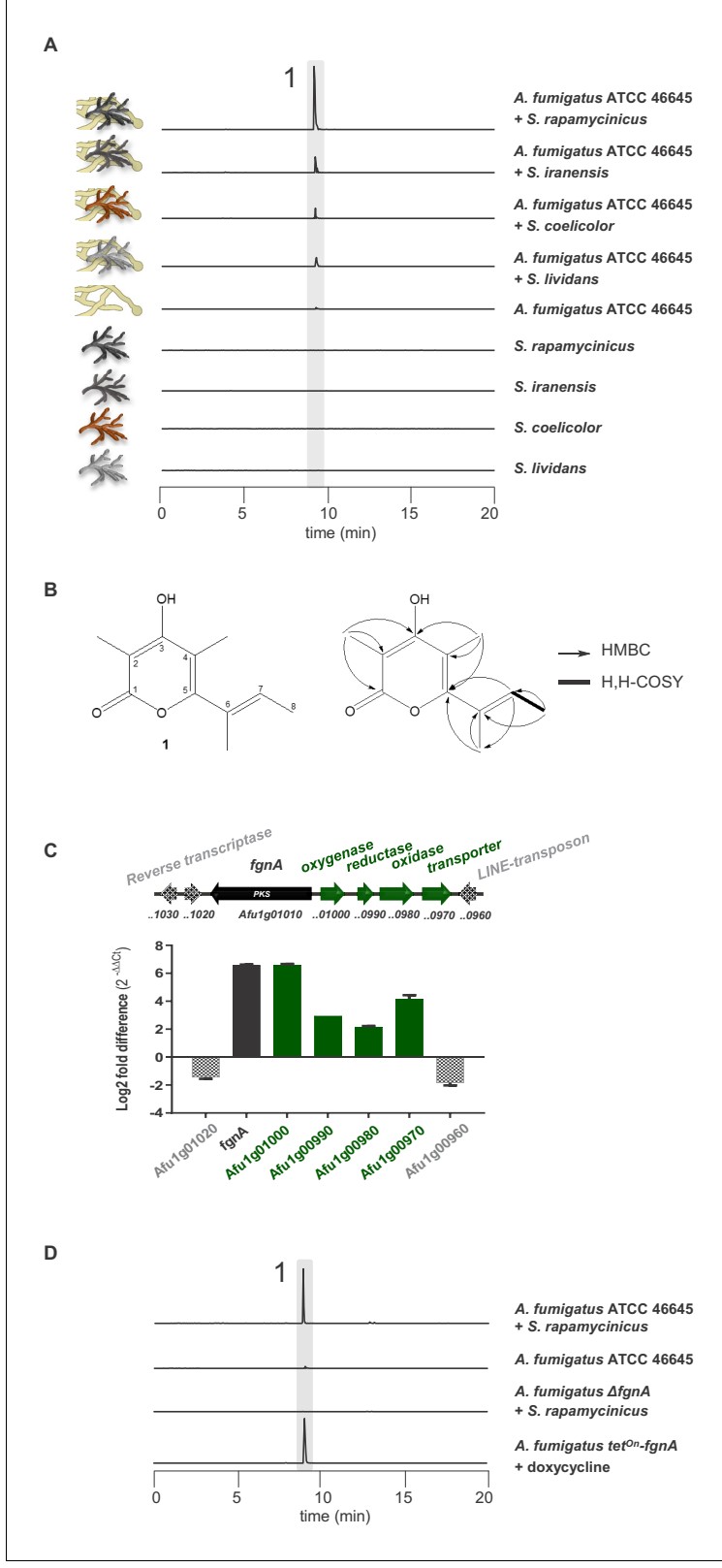

**Figure 1.** Induction of SM production in *A. fumigatus* by *Streptomyces* spp. and identification of the gene responsible for the SM production. (**A**) LC-MS analysis of supernatants of *A. fumigatus* strains after 12 h in axenic or in co-culture with indicated *Streptomyces* species showing EIC traces for *m/z* 195 [M+H]⁺, corresponding to the newly formed compound 1, which is indicated by the highlighted strip. (**B**) Structure and 2D-NMR correlations of *Figure 1 continued on next page*

*Figure 1 continued*

compound 1, fumigermin. (C) Transcription analysis of the *A. fumigatus* ATCC 46645 *fgn* cluster and adjacent genes determined by qRT-PCR after 5 h of co-cultivation with *S. rapamycinicus*. Relative mRNA levels were compared to the β-actin gene transcript levels. The relative expression values are visualized as relative log₂ fold difference ($2^{-\Delta\Delta Ct}$) of transcript levels for each gene in co-culture with *S. rapamycinicus* compared to transcript levels of the respective gene in *A. fumigatus* ATCC 46645 in axenic culture. Data are representative of three biological replicates with three technical replicates. Error bars indicate standard error of the mean. (D) EIC traces of supernatants of the different *A. fumigatus* strains in co-cultivation with *S. rapamycinicus* or under inducing conditions are shown for $m/z$ 195 [M+H]⁺, which corresponds to fumigermin (1).

The online version of this article includes the following source data and figure supplement(s) for figure 1:

**Source data 1.** Transcriptional analysis of the *fgn* cluster.
**Figure supplement 1.** NMR spectra of fumigermin.
**Figure supplement 2.** NMR spectra of fumigermin.
**Figure supplement 3.** NMR spectra of fumigermin.
**Figure supplement 4.** Structure of fumigermin (1).
**Figure supplement 5.** Verification of fungal transformant strains by Southern blot.
**Figure supplement 6.** Verification of inducibility of fumigermin biosynthesis by *S. rapamycinicus* supernatant in *A. fumigatus* using LC-MS metabolic analysis.

---

Additionally, the structure of the compound suggested that its biosynthesis is catalyzed by a partially reducing polyketide synthase (PKS). Bioinformatic inspection of the genome (*Bignell et al., 2016*) indicated that only a few clusters matched this prediction. One of these clusters, whose products are so far unknown, was the uncharacterized gene cluster *Afu1g00970-Afu1g01010* (*Supplementary file 1*), containing the putative PKS gene *Afu1g01010*.

To verify whether the selected genes are involved in biosynthesis of the detected metabolite produced in the bacterial-fungal interaction, we analyzed expression of cluster genes *Afu1g00970-Afu1g01010* of *A. fumigatus* and their 3' and 5' adjacent genes by qRT-PCR after 5 h of co-culture. As indicated in *Figure 1C*, the steady-state mRNA levels of the putative cluster genes of *A. fumigatus* ATCC 46645, including gene *Afu1g01010*, were markedly upregulated during co-cultivation of *A. fumigatus* ATCC 46645 with *S. rapamycinicus*. To unequivocally prove that the identified PKS gene *Afu1g01010* encodes the key enzyme involved in the biosynthesis of fumigermin, we deleted *Afu1g01010* in the genome of *A. fumigatus* ATCC 46645 (*Figure 1—figure supplement 5*). LC-MS monitoring of the co-cultivation of the resulting deletion mutant with *S. rapamycinicus* demonstrated that production of fumigermin (1) was fully abolished (*Figure 1D*), confirming that the PKS gene *Afu1g01010* is involved in its biosynthesis. This was further proven by overexpressing *Afu1g01010*, hereby named *fgnA*, using the inducible *tet*^*On* promoter system in *A. fumigatus* ATCC 46645, which led to the appearance of fumigermin under inducing conditions in the absence of bacteria (*Figure 1D*). Taken together, these data clearly prove that the identified *fgnA* gene is strictly required for biosynthesis of the novel metabolite fumigermin (1).

## The *fgnA* polyketide synthase gene is essential for production of fumigermin

To characterize the biosynthetic pathway of fumigermin, we expressed the *fgnA* gene from strain ATCC 46645 heterologously in *A. nidulans,* under control of the constitutive *gpdA* promoter. Metabolic profiling of the culture of *A. nidulans* overexpressing the *fgnA* gene of strain ATCC 46645 revealed the formation of 1, a metabolite identical to fumigermin which had been previously detected during the co-cultivation of *A. fumigatus* and *S. rapamycinicus* (*Figure 2A*). This finding suggested that fumigermin, that is compound 1, is a direct product of the encoded polyketide synthase FgnA of *A. fumigatus*.

To prove that the encoded putative tailoring enzymes are non-essential for biosynthesis of fumigermin, we transferred the entire *fgn* cluster from *A. fumigatus* ATCC 46645 to *A. nidulans* RMS011. For this purpose, we applied our previously developed heterologous expression system allowing for expression of several genes from a polycistronic mRNA in a eukaryotic host (*Unkles et al., 2014*; *Hoefgen et al., 2018*). This expression system is based on post-translational cleavage of proteins at the viral 2A peptide sequences (*Whitton et al., 2005*). Briefly, all cluster genes were cloned in a

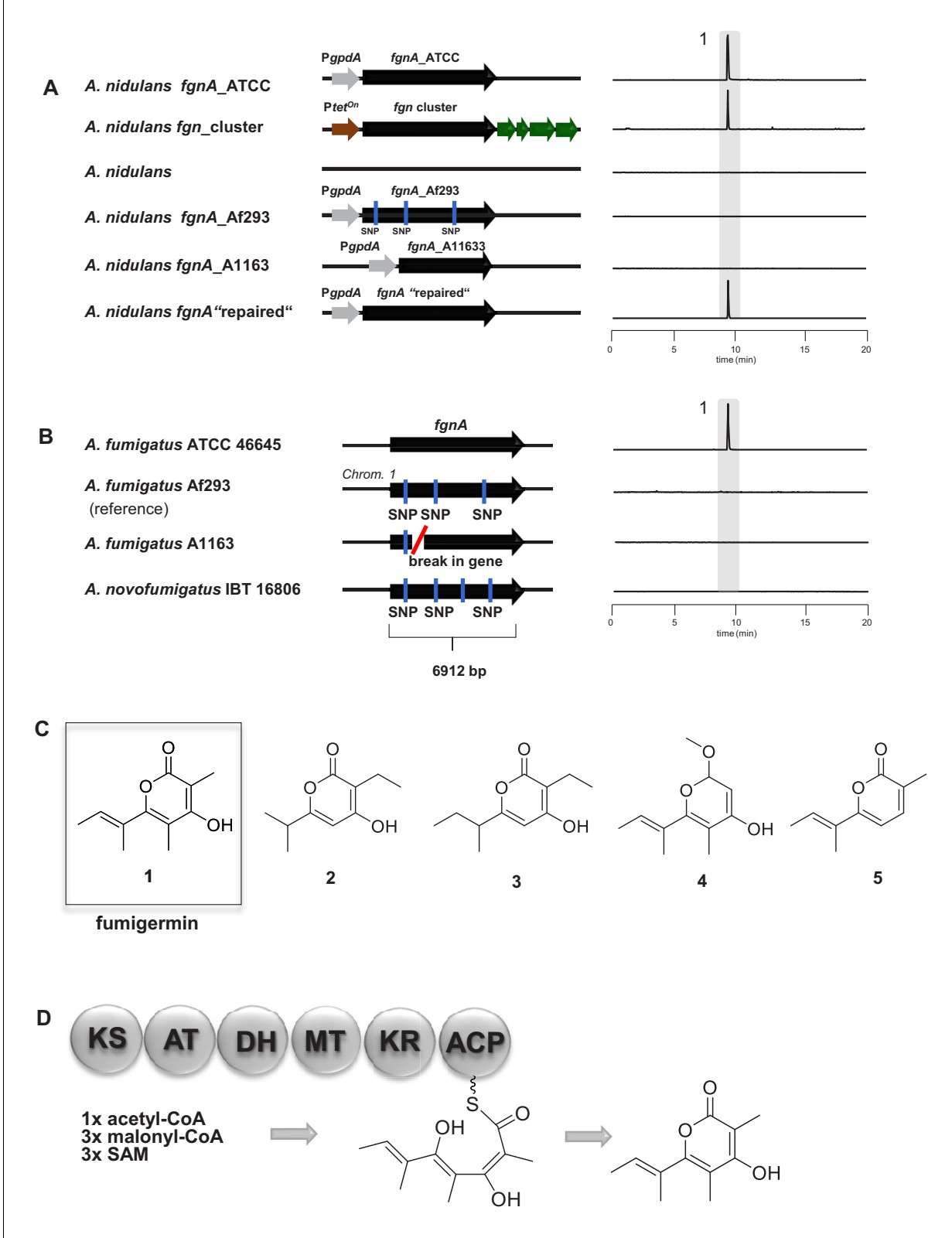

**Figure 2.** Molecular analysis of the *fgnA* PKS gene and structure of its biosynthesized product fumigermin. (A) Heterologous expression of *fgnA* genes from the indicated *A. fumigatus* strains in the host *A. nidulans*. The entire *fgn* cluster was transferred and expressed in the host *A. nidulans*. A 'repaired' *fgnA* based on the gene of A1163 was expressed in *A. nidulans*. (B) PKS gene *fgnA* in different indicated *A. fumigatus* strains. SNPs and a gene break are marked. EIC traces of supernatants of the different *A. fumigatus* strains in co-cultivation with *S. rapamycinicus* are shown in (A) and (B) for *m/z* 195

*Figure 2 continued on next page*

*Figure 2 continued*

[M+H]$^+$, which corresponds to fumigermin (1). (**C**) Comparison of the discovered α-pyrone polyketide compound fumigermin (1) and the structurally related bacterial pyrones germicidin A (2), germicidin B (3), nectriapyrone (4) and gibepyrone (5). (**D**) FgnA enzyme domains and proposed biosynthesis of fumigermin.

The online version of this article includes the following source data and figure supplement(s) for figure 2:

**Figure supplement 1.** Expression of the *A. fumigatus fgn* cluster in the heterologous host *A. nidulans*.

**Figure supplement 1—source data 1.** Global proteome of *A. nidulans fgn*_cluster.

**Figure supplement 2.** Verification of inducibility of fumigermin biosynthesis by *S. rapamycinicus* in a heterologous host context using LC-MS metabolic analysis.

**Figure supplement 3.** PCR strategy to locate the breakage of gene *fgnA* from *A. fumigatus* A1163.

**Figure supplement 4.** Comparison of the biosyntheses of fungal α-pyrones.

plasmid such that they were all separated by 2A peptide-encoding sequences. To facilitate selection of transformants harboring such plasmids, we employed a gene encoding a split-Venus fluorescent protein that, in addition, contained a nuclear localization sequence. In the plasmid, the gene cluster was flanked by two genes, each encoding a half of the Venus gene. Consequently, when all genes of the plasmid are expressed, the two Venus halves (N+C) are also produced and bind to each other to form a functional Venus fluorescent protein. As a result, fluorescence in the fungal nucleus indicates correct translation of the encoded proteins. The entire cluster construct was placed under the control of the *tet$^{On}$* promoter, allowing inducible expression of all genes from a polycistronic mRNA (*Figure 2—figure supplement 1A*). Although all cluster proteins were produced in the heterologous strain, as evidenced by the fluorescent signal localized to the nuclei (*Figure 2—figure supplement 1B*) and additionally supported by label-free proteomic analysis (*Figure 2—figure supplement 1C*), no further compound, other than fumigermin, was detected in the heterologous expression strain (*Figure 2A*, *Figure 2—figure supplement 1B*). This result indicates that the additional enzymes do not modify the chemical product of FgnA, even when co-expressed with the functional FgnA PKS.

## The partially reducing FgnA synthesizes fumigermin *via* a PKS I-derived pathway

The deduced amino acid sequence of FgnA analyzed by CD-search (*Marchler-Bauer and Bryant, 2004*) revealed the presence of a ketosynthase (KS), acyltransferase (AT), dehydratase (DH), methyl transferase (MT), ketoreductase (KR) and an acyl-carrier protein (ACP) domain, classifying this enzyme as a partially reducing PKS (*Cox, 2007*). Isotope labeling studies using (2-$^{13}$C)acetate and (*methyl*-$^{13}$C)methionine showed that eight carbon atoms of fumigermin originate from acetate, while the remaining three carbons derived from S-adenosyl methionine, agreeing with the existence of an MT domain in the PKS (*Figure 1—figure supplement 4*). The conserved domains together with the precursor studies led to the proposed model for fumigermin biosynthesis by FgnA (*Figure 2D*).

## In *A. fumigatus* a functional FgnA is restricted to strain ATCC 46645

Among *Aspergillus* spp., the *fgn* cluster is unique to *A. fumigatus* and the closely related species *A. novofumigatus* (*Figure 2B*). Moreover, the cluster is rarely found in the different sequenced *A. fumigatus* genomes, which is consistent with a previously reported analysis of clusters in various *A. fumigatus* isolates (*Lind et al., 2017*). Amongst the *A. fumigatus* strains harboring the *fgn* cluster, production of fumigermin was restricted to strain ATCC 46645. We could not detect fumigermin in a co-culture of *S. rapamycinicus* with other *A. fumigatus* strains, such as Af293 and A1163, or *A. novofumigatus* IBT 16806 (*Figure 2B*). To determine the reason for the missing compound production in strains Af293, A1163 and IBT 16806, we analyzed their respective genome sequences retrieved from the NCBI database, with a focus on the *fgnA* gene. For strain A1163, the available genome sequence did not cover the area around the putative *Afu1g01010* cluster. Thus, we amplified the *fgnA* gene of A1163 by PCR and sequenced it. As indicated by the gene sequences, the corresponding *fgnA* gene of strains Af293 and IBT 16806 contained several missense SNPs, while that of strain A1163 had undergone a chromosomal breakage around 800 bp from the start codon of *fgnA*, and contained a C → T missense mutation at position 272 (*Figure 2B*, *Figure 2—figure supplement 3*). Thus, these genes appeared to be non-functional.

To further verify the assumption that the *fgnA* genes are not functional in several *A. fumigatus* strains, we overexpressed the *fgnA* genes of both the Af293 and A1163 strains in *A. nidulans*. As predicted, neither FgnA PKS was able to produce fumigermin (*Figure 2A*). To test whether the repair of the SNP in a A1163-derived PKS gene would rescue the function of the biosynthetic enzyme, the two PCR-derived DNA fragments covering the A1163 *fgnA* gene were first ligated in a plasmid and then the missense SNP was corrected. The heterologous overexpression of the resulting 'repaired' *fgnA* gene in *A. nidulans* under the control of the *gpdA* promoter led to formation of the expected polyketide metabolite fumigermin, confirming that this reconstruction functionally rescued the 'inactive' A1163 *fgnA* gene and led to production of the previously observed metabolite derived from the *fgnA* gene (*Figure 2A*).

## The fumigermin synthase gene is spread among distantly related fungal classes

To identify potential orthologs of FgnA and to understand its relation to other fungal PKS, we performed Uniprot database searches and phylogenetic analyses. The obtained data indicate that additional orthologs of the entire *fgn* cluster are found in distantly related fungi such as the major pathogen of wheat *Parastagonopora nodorum* from the fungal taxon Dothideomycetes (*Hane et al., 2007*), the bat pathogen *Pseudogymnoascus pannorum* which belongs to the Leotiomycetes (*Minnis and Lindner, 2013*) and fungi of the Sodariomycetes such as the parasitic microfungus *Escovopsis weberi* (*Reynolds and Currie, 2004*) and the endophyte *Hypoxylon* sp. (*Hsieh et al., 2005*; *Supplementary file 5*). As these microorganisms are derived from different fungal classes and occupy diverse habitats, it is conceivable that the *fgn* cluster was spread between different fungal species through horizontal gene transfer.

Surprisingly, a similarity search based on the protein sequence of FgnA did not retrieve the stand-alone PKS enzymes Nec1 and Gpy1, which catalyze the biosynthesis of the structurally similar microbial SMs nectriapyrone (*Figure 2—figure supplement 4*; *Janevska et al., 2016*) and gibepyrone (*Motoyama et al., 2019*), respectively. Despite also being classified as fungal iterative type I PKS, Nec1 and Gpy1 are not orthologs of FgnA, and show low similarity to FgnA at the amino acid level. Moreover, Nec1 and Gpy1 also constitute interesting examples of stand-alone PKS enzymes that produce compounds similar to fumigermin and that do not require additional cluster-encoded enzymes.

## Fumigermin reversibly inhibits the germination of spores of the inducing *S. rapamycinicus*

Fumigermin displayed no remarkable antimicrobial effects, except for a modest inhibitory activity against MRSA (data not shown). To learn more about the potential ecological role of fumigermin, we investigated its effects on the inducing streptomycetes. Fumigermin is structurally related to germicidins, which have been shown to function as auto-regulators of spore germination, that is they inhibit the germination of the producing streptomycetes (*Aoki et al., 2011*). We therefore tested whether fumigermin has a similar activity, which would provide an explanation for the interaction between *A. fumigatus* and the streptomycetes. As shown in *Figure 3A and B*, the number of germinated *S. rapamycinicus* spores was drastically reduced by fumigermin in a dose-dependent manner. A comparison of fumigermin and germicidin B demonstrated that both compounds showed the same anti-germination effect against spores of *S. rapamycinicus* (*Figure 3C*). Moreover, fumigermin also inhibited germination of *S. rapamyinicus* spores out of solid medium (*Figure 3D*). To reveal whether the inhibition of spore germination was reversible, we incubated *S. rapamycinicus* spores with fumigermin for 90 h, and subsequently washed and re-cultivated the spores in fresh media. As shown in *Figure 3E* and *Figure 3—figure supplement 1*, the spores' ability to germinate and form a mycelial network was regained. Thus, fumigermin is a reversible germination inhibitor, which keeps *S. rapamycinicus* spores in a dormant state and leaves them ungerminated as long as the compound is present. Notably, fumigermin could be re-purified from 90 h-old supernatant of a fumigermin-treated *S. rapamycinicus* spore sample, and further re-used to inhibit germination of fresh *S. rapamycinicus* spores (*Figure 3—figure supplement 1*). This finding points towards high stability of the compound.

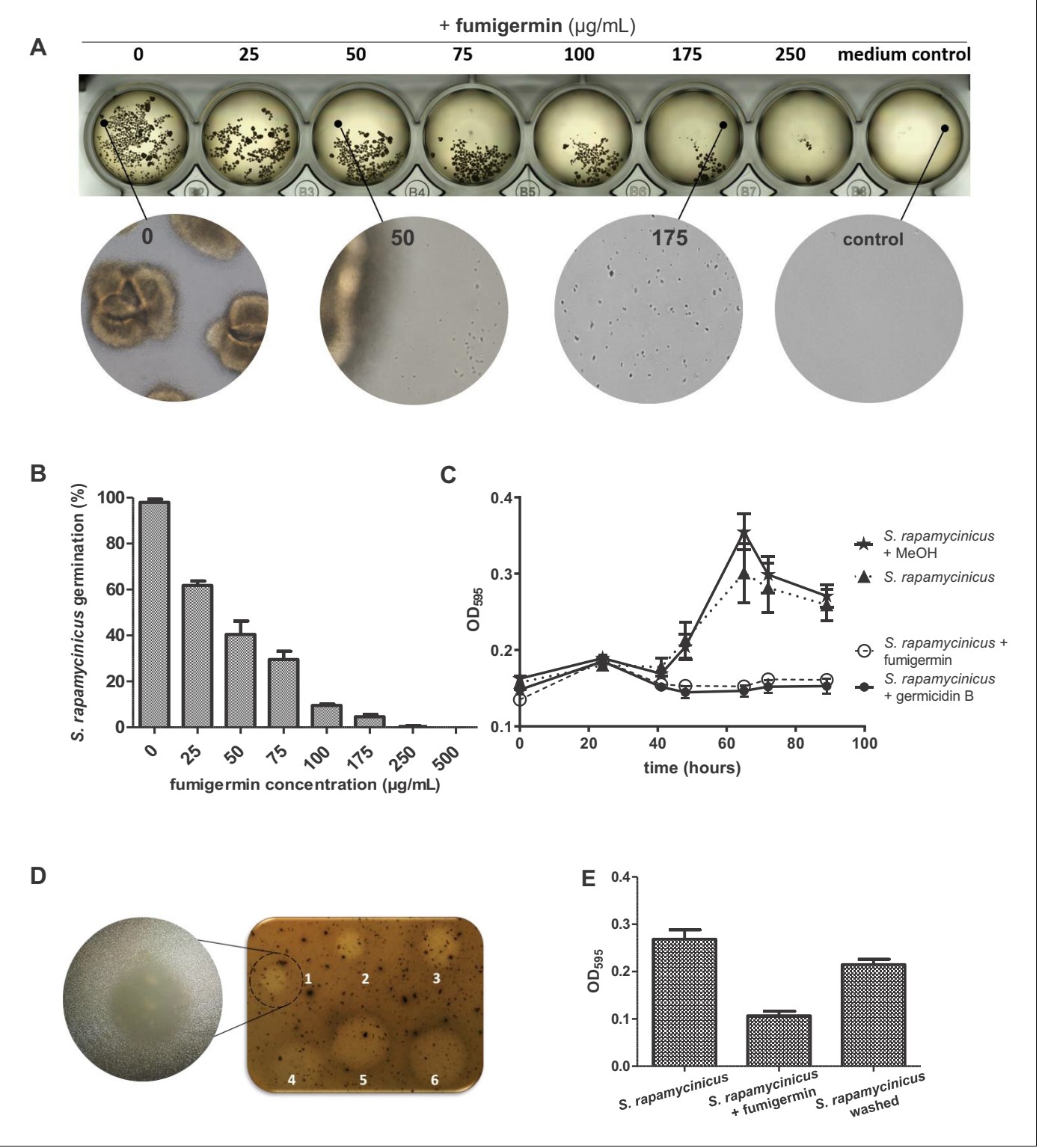

**Figure 3.** Activity of fumigermin against germination of *S. rapamycinicus* spores. (**A**) $10^7$ spores of *S. rapamycinicus* were inoculated in 500 µL M79 medium in a 48-well plate and treated with increasing fumigermin concentrations. Image taken after 5 days of growth at 28°C, 500 rpm. Magnified areas zoom in on an untreated sample, sample treated with 50 µg/mL or 175 µg/mL fumigermin and a control sample containing medium with no spores. (**B**) $OD_{595}$ measurements were taken for each well using a Tecan device. The mean $OD_{595}$ value of the untreated *S. rapamycinicus* samples was set as 100% germination, and used to calculate the relative germination percentage of the fumigermin-treated samples. (**C**) Graph of 90-h-long monitoring of *S. rapamycinicus* spore germination inhibition at 28°C using germicidin B and fumigermin (250 µg/mL). Controls were treated with MeOH or remained

*Figure 3 continued on next page*

*Figure 3 continued*

untreated. An increase in the OD$_{595}$ value is indicative of *S. rapamycinicus* spore germination. Data in (B, C) correspond to three biological samples, each with three technical replicates. Error bars indicate standard error of the mean. (D) Qualitative inhibition of formation of *S. rapamycinicus* aerial mycelia by fumigermin. Oatmeal agar plates with embedded *S. rapamycinicus* spores were point-inoculated with different concentrations of fumigermin: (1) 12.5 µg, (2) 25 µg, (3) 50 µg, (4) 75 µg, (5) 100 µg, (6) 250 µg and incubated for 1 week at 28°C. Left, zoom of area (1). (E) Reversibility of germination inhibition. *S. rapamycinicus* spores were incubated with 250 µg/mL fumigermin. After 5 days, the spores did not germinate; when recovered, washed and placed in fresh medium, the spores grew and formed mycelia.

The online version of this article includes the following source data and figure supplement(s) for figure 3:

**Source data 1.** OD$_{595}$ measurement of *S. rapamycinicus* growth.
**Figure supplement 1.** Stability of fumigermin and reversibility of its inhibitory action.
**Figure supplement 2.** Germination assay of *S. lividans* and *S. coelicolor* treated with fumigermin.
**Figure supplement 3.** *S. rapamycinicus* cannot degrade fumigermin (top), but *Pseudomonas aeruginosa* can (bottom).

Because *S. lividans* and *S. coelicolor* only triggered production of moderate fumigermin titers, we tested whether fumigermin also affected the growth of these streptomycetes. The germination of *S. coelicolor* and *S. lividans* was only slightly inhibited when treated with fumigermin. Moderate germination inhibition was observed for *S. coelicolor* treated with the rather high concentration of 1000 µg/mL fumigermin (*Figure 3—figure supplement 2*). The inhibitory effect was lost after 3 days of incubation with the compound, suggesting that fumigermin might be degraded by this *Streptomyces* spp. To test this assumption, the purified metabolite was supplemented to growing cultures of *S. rapamyinicus*, *S. iranensis*, *S. lividans* and *S. coelicolor*, and a control culture of *Pseudomonas aeruginosa* PAO1. After 24 h of incubation, the levels of fumigermin were determined by LC-MS (*Figure 3—figure supplement 3*). The analysis showed that none of the streptomycetes tested was able to degrade fumigermin, which was in contrast to the *P. aeruginosa* PAO1 control culture where fumigermin was almost depleted in the culture supernatant. As expected, fumigermin had no obvious effect on the fitness of the producing isolate *A. fumigatus* ATCC 46645 at least when cultivated in monoculture (data not shown).

## Discussion

### Fumigermin reversibly inhibits germination of susceptible *Streptomyces* strains

Employing interactions of microorganisms to activate cryptic SM gene clusters has been successfully established as a methodology to identify novel compounds (*Netzker et al., 2018*). Here, we uncovered a novel fungal SM, production of which was induced during the interaction of *A. fumigatus* with *S. rapamycinicus*. It is reasonable to assume that SMs are produced for the benefit of the producer microorganism in a specific niche (*Macheleidt et al., 2016*). However, the ecological benefit for the producer has only been established for a few cases (*Notz et al., 2002*; *Tanaka et al., 2005*; *Khalil et al., 2019*). For instance, *A. flavus* produces aflavinines as protection against the beetle *Carpophilus hemipterus* (*Gloer and Truckenbrod, 1988*). The fungal endophyte *Epichloë festucae* produces peramine to protect its plant host from feeding by *Listronotus bonariensis* (*Tanaka et al., 2005*). Moreover, fusaric acid produced by *Fusarium graminearum* represses production of an antifungal compound in *Pseudomonas fluorescens* CHA0 (*Notz et al., 2002*). Consistently, fumigermin is not produced in an axenic *A. fumigatus* ATCC 46645 culture, but is produced in high amounts and excreted by *A. fumigatus* ATCC 46645 when co-cultured with *S. rapamycinicus*.

*Aspergillus* spp., in particular *A. fumigatus,* and *Streptomyces* spp. have been isolated previously from a soil sample and found to share the same habitat (*Alani et al., 2012*; *Khalil et al., 2019*), making it likely that this laboratory scenario is also relevant for the ecological niche of the analyzed microorganisms. However, there is little insight into the ecology of bacterial-fungal interactions yet. In the case of *A. fumigatus* ATCC 46645 and *S. rapamycinicus* reported here, the two microorganisms are apparently interacting in an antagonistic manner, because the fungus is triggered by the bacterium to produce the germination inhibitor fumigermin to protect its habitat's resources (*Figure 4*). As fumigermin's inhibitory activity is reversible, it is likely that the fungus does not intend to kill the bacterium.

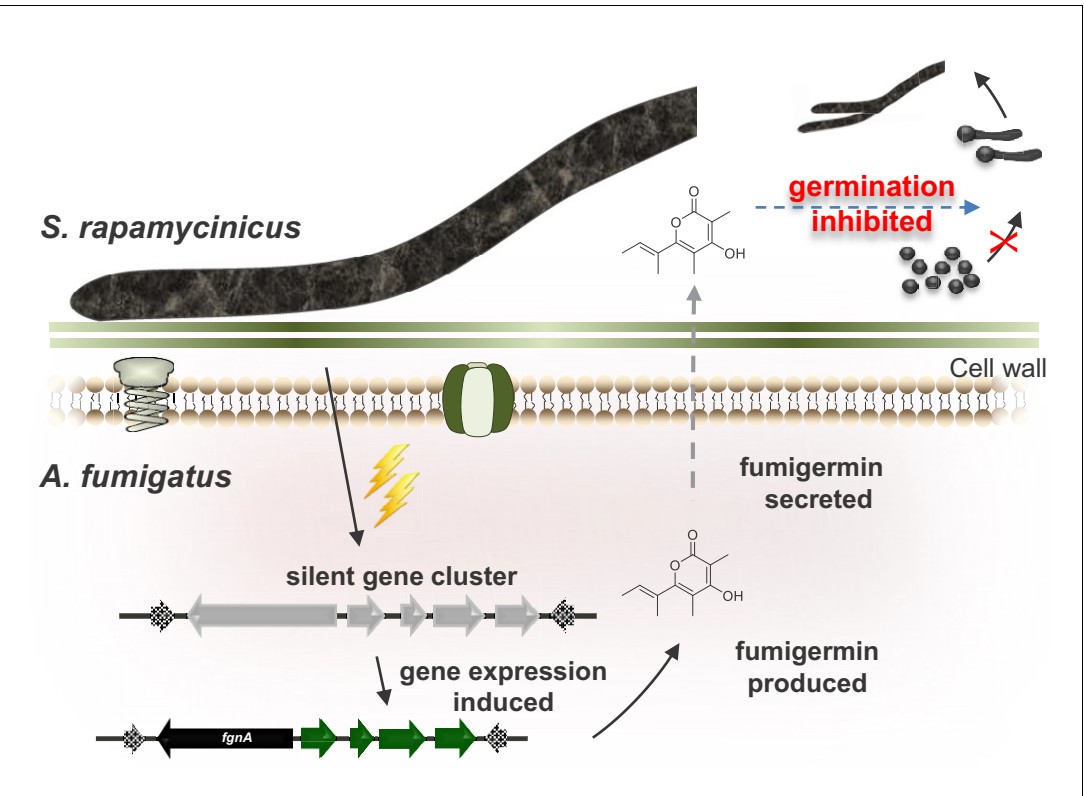

**Figure 4.** Model of the interaction between *S. rapamycinicus* and *A. fumigatus* ATCC 46645, which results in production of fumigermin. During the co-cultivation of the two microorganisms, the bacterium *S. rapamycinicus* induces expression of the *fgn* gene cluster of the fungus *A. fumigatus* ATCC 46645, which leads to production of the SM fumigermin. Fumigermin is secreted and further inhibits germination of *S. rapamycinicus* spores.

In this work, several *Streptomyces* spp. were found to induce fumigermin biosynthesis. Whereas *S. iranensis* is phylogenetically closely related to the fumigermin-inducer *S. rapamycinicus*, *S. coelicolor* and *S. lividans* are rather distantly related to *S. rapamycinicus* (*Nouioui et al., 2018*). This phylogenetic relationship mirrors the extent of induction of fumigermin production in *A. fumigatus* and also correlates with the activity of fumigermin against the inducing *Streptomyces* spp.. More specifically, fumigermin was highly active against the main fumigermin inducers *S. rapamycinicus* and its close relative *S. iranensis* (data not shown). On the other hand, *S. coelicolor* and *S. lividans* triggered production of lower levels of fumigermin and were not susceptible to fumigermin. Our data on the degree of induction and the susceptibility of *Streptomyces* species thus indicate the specificity of the interaction of *S. rapamycinicus*/*S. iranensis* and *A. fumigatus*. *S. coelicolor* and *S. lividans* are known producers of the structurally related α-pyrone metabolites germicidins (*Song et al., 2006*; *Chemler et al., 2012*). It is thus likely that these streptomycetes harbor an intrinsic resistance mechanism against germination-inhibiting α-pyrone compounds. Because none of the *Streptomyces* species analyzed here degraded fumigermin, it is possible that other resistance mechanisms such as specific export of germicidins, are responsible for the intrinsic resistance.

Germicidins are autoregulatory inhibitors of germination of dormant spores in a *Streptomyces* population. *Streptomyces* germ tubes secrete germicidins, which presumably block the activity of a membrane located Ca$^{2+}$-dependent ATPase (*Eaton and Ensign, 1980*; *Grund and Ensign, 1985*; *Aoki et al., 2011*). The role of germicidins for the ecology of streptomycetes is a matter of debate (*Čihák et al., 2017*). It is conceivable that *A. fumigatus* has acquired the ability to biosynthesize a functionally similar compound and is exploiting this effect to its advantage during interaction with the bacteria, as means of protection of the habitat's resources from competitor microorganisms.

## The biosynthesis routes of α-pyrone-based germination inhibitors possibly developed by convergent evolution

The structural similarity between fumigermin and germicidins and their same biological activity initially suggested that the compounds share a common biosynthetic origin. However, germicidins and fumigermin are products of pathways that are different. Whereas germicidins are produced by a type II fatty acid synthase coupled with a bacterial type III PKS that uses precursors derived from fatty acid biosynthesis (*Figure 5* and *Chemler et al., 2012*), fumigermin biosynthesis involves a fungal iterative type I PKS. This finding makes a horizontal gene transfer from bacteria to fungi unlikely and rather points to a possible case of convergent evolution of the same lead structure in the bacterial and fungal kingdom.

## A. fumigatus-specific fumigermin PKS gene is sufficient for germicidin production despite being located in a cluster

It is remarkable that the five cluster genes including *fgnA* are conserved in distantly related fungi, supporting the assumption of a *fgn* cluster. It is thus possible that originally all five genes were functional, but that the genes encoding tailoring enzymes became inactive from mutations, similar to the case of the *fgnA* gene in strains A1163 and Af293. In *A. fumigatus*, the *fgn* locus has been shown to be particularly prone to rearrangements and breakages, possibly as a result of being flanked by potential transposons (*Figure 1C*, *Supplementary file 1*). Consistently, Lind et al. have shown that the cluster 'jumped' to different genomic locations in three of the six analyzed cluster-containing strains, in one case being located inside the fumagillin/pseurotin supercluster (*Wiemann et al., 2013*; *Lind et al., 2017*). Additionally, loss of gene clusters apparently tends to occur at higher rates when they are located near telomere ends (*Young et al., 2015*), which might explain that the *fgn* cluster is only present in a few *A. fumigatus* isolates.

Collectively, a rational explanation regarding the fate of the *fgn* cluster is that the modifications suffered as a consequence of its chromosomal location, or because of repeated transpositions driven by the putative cluster-flanking mobile elements, have led to breakages and SNP formation and a consequent loss of cluster functionality in most of the *A. fumigatus* strains. Our data so far indicate that the isolate ATCC 46645 is the only *Aspergillus* sp. in which the cluster was maintained, even though the tailoring enzymes are apparently non-functional (*Figure 2—figure supplement 1*). The production of fumigermin likely supported the conservation of FgnA, as it was advantageous for the particular isolate in its interaction with streptomycetes (*Figure 4*).

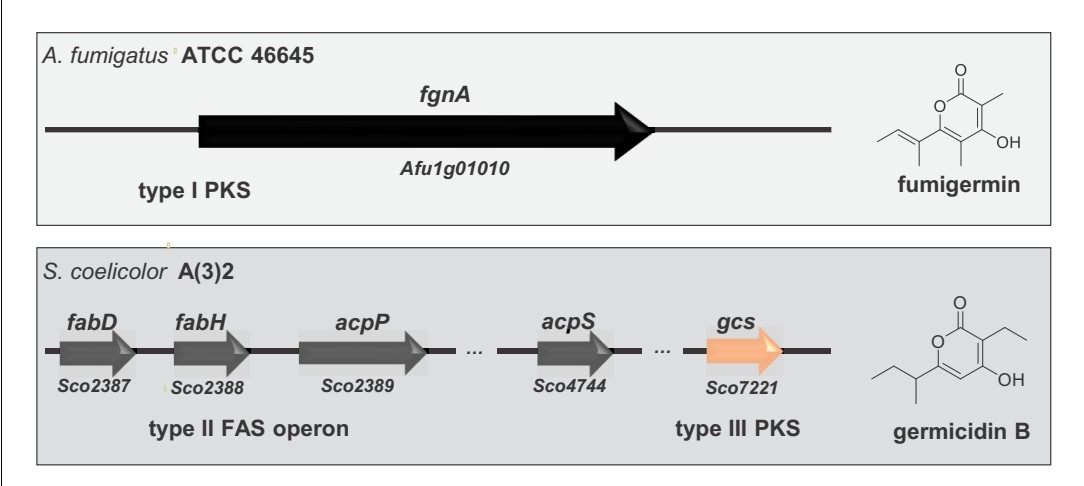

**Figure 5.** Alignment of genes involved in biosynthesis of the fungal compound fumigermin and the bacterial compound germicidin B. Germicidin synthase is a type III bacterial PKS, while FgnA is a fungal iterative type I PKS. The synthesis of germicidins is coupled with fatty acid biosynthesis in *S. coelicolor* - the branched precursors are synthesized by FabD and FabH, after which they are further loaded onto and modified by Gcs (*Chemler et al., 2012*). In case of *A. fumigatus*, FgnA catalyzes the condensation of malonyl-CoA units to an acetyl-CoA starter unit independently of the fatty acid biosynthesis pathway. There is no striking similarity between Gcs and FgnA.

# Materials and methods

## Key resources table

| Reagent type (species) or resource | Designation | Source or reference | Identifiers | Additional information |
|---|---|---|---|---|
| Strain, strain background (*Aspergillus fumigatus*) | Af293 | *Nierman et al., 2005* | | Wild-type strain, *MAT1-2* |
| Strain, strain background (*Aspergillus fumigatus*) | A1163 | *da Silva Ferreira et al., 2006* | | Wild-type strain, *MAT1-1* |
| Strain, strain background (*Aspergillus fumigatus*) | ATCC 46645 | *Langfelder et al., 1998* | | Wild-type strain, *MAT1-1* |
| Strain, strain background (*A. novofumigatus*) | IBT 16806 | *Hong et al., 2005* | | Wild-type strain |
| Strain, strain background (*Aspergillus fumigatus*) | ΔfgnA | This paper | | ATCC 46645 background, *ptrA::fgnA*, PT$^R$ |
| Strain, strain background (*Aspergillus fumigatus*) | tet$^{On}$-fgnA | This paper | | ATCC 46645 background, *tet$^{On}$-fgnA*, PT$^R$ |
| Strain, strain background (*Aspergillus nidulans*) | RMS011 | *Stringer et al., 1991* | | *pabaA1, yA2; ΔargB ::trpCΔB, trpC801, veA1* |
| Strain, strain background (*Aspergillus nidulans*) | fgnA_ATCC | This paper | | RMS011 background, pUC18-*gpdA*-fgnA_ATCC, ArgB2$^+$ |
| Strain, strain background (*Aspergillus nidulans*) | fgnA_Af293 | This paper | | RMS011 background, pUC18-*gpdA*-fgnA_Af293, ArgB2$^+$ |
| Strain, strain background (*Aspergillus nidulans*) | fgnA_A1163_repaired | This paper | | RMS011 background, pUC18-*gpdA*-fgnA_A1163*, ArgB2$^+$ |
| Strain, strain background (*Aspergillus nidulans*) | fgn_cluster | This paper | | RMS011 background, fgnA,Afu1g01000,Afu1g00990,Afu1g00980,Afu1g00970, ArgB2$^+$ |
| Strain, strain background (*Streptomyces rapamycinicus*) | ATCC 29253 | *Kumar and Goodfellow, 2008* | | Wild-type strain |
| Strain, strain background (*Streptomyces iranensis*) | HM35 | *Hamedi et al., 2010* | | Wild-type strain |
| Strain, strain background (*Streptomyces lividans*) | PM02 | *van Dissel et al., 2015* | | Wild-type strain |
| Strain, strain background (*Streptomyces coelicolor*) | A(3)2 | *Erikson, 1955* | | Wild-type strain |
| Commercial assay or kit | NEB HiFi | New England Biolabs, Frankfurt, Germany | E2621L | |
| Commercial assay or kit | Universal RNA Purification Kit | Roboklon, Berlin, Germany | E3598 | |
| Chemical compound, drug | germicidin B | Cayman Chemical, Ann Arbor, USA | CAS number 150973-78-7 | |

*Continued on next page*

*Continued*

| Reagent type (species) or resource | Designation | Source or reference | Identifiers | Additional information |
|---|---|---|---|---|
| Chemical compound, drug | digoxigenin-11- dUT | Jena BioScience, Jena, Germany | Catalog number NU-803-DIGXS | |
| Software, algorithm | GraphPad Prism 5 | GraphPad Software Inc,La Jolla, USA | RRID:SCR_002798 | |
| Software, algorithm | Thermo Xcalibur Qual Browser | Thermo Scientific, Dreieich, Germany | | |
| Software, algorithm | Proteome Discoverer (PD) 2.2 | Thermo Scientific, Dreieich, Germany | RRID:SCR_014477 | |
| Software, algorithm | Shimadzu Class-VP software (version 6.14 SP1) | Shimadzu, Duisburg, Germany | 6.14 SP1 | |
| Other | Hoechst 34580 | Thermo Fischer Scientific, Dreieich, Germany | H21486 | |

## Microorganisms, media and cultivation

Microorganisms used in this study are listed in the Key Resources Table. *A. fumigatus* and *A. nidulans* strains were cultivated in *Aspergillus* minimal medium (AMM) at 37°C, 200 rpm (*Brakhage and Van den Brulle, 1995*). Fungal pre-cultures were inoculated with $4 \times 10^8$ spores per mL. For *A. fumigatus* $tet^{On}$-fgnA and *A. nidulans* fgn_cluster, 10 µg/mL doxycycline was used to induce the $tet^{On}$ inducible system. For *A. nidulans* strains, the culture medium was supplemented with arginine (871 µg/mL) and *p*-aminobenzoic acid (3 mg/L) when needed. *S. rapamycinicus* and *S. coelicolor* were cultivated on oatmeal agar plates (*Shirling and Gottlieb, 1966*) for sporulation and in M79 medium (*Prauser and Falta, 1968*) for liquid cultivation. *S. lividans* was cultivated on mannitol soya flour (MS) agar plates (*Hobbs et al., 1989*) to obtain spores. For cultivation in liquid media tryptone soy broth (TSB) was employed (*Kieser et al., 2000*) with 10% (w/v) sucrose. Cultures were propagated at 28°C and shaken at 160 rpm. For co-cultivation experiments, mycelia of *A. fumigatus* (~16 h old) were separated from the medium using Miracloth (Merck Millipore, Darmstadt, Germany), placed in fresh AMM and inoculated with 1/20 vol of the streptomycete culture as previously described (*Schroeckh et al., 2009*), or with 1/20 vol of the streptomycete culture supernatant, filtered through a 0.22 µm filter. RNA extraction for expression analysis was performed after 5 h of co-cultivation, while samples for LC-MS analysis were taken after 12 h of co-cultivation.

## Extraction of fungal compounds and LC-MS analysis

The culture broth containing fungal mycelium with and without bacteria was homogenized using an ULTRA-TURRAX (IKA-Werke, Staufen, Germany). Homogenized cultures were extracted twice with a total of 100 mL ethyl acetate, dried with sodium sulfate and concentrated under reduced pressure. For LC-MS analysis, the dried extracts were dissolved in 1 mL of methanol and loaded onto an ultra-high-performance liquid chromatography (LC)–MS system consisting of an UltiMate 3000 binary rapid-separation liquid chromatograph with photodiode array detector (Thermo Fisher Scientific, Dreieich, Germany) and an LTQ XL linear ion trap mass spectrometer (Thermo Fisher Scientific, Dreieich, Germany) equipped with an electrospray ion source. The extracts (injection volume, 10 µl) were analyzed on a 150 mm by 4.6 mm Accucore reversed-phase (RP)-MS column with a particle size of 2.6 µm (Thermo Fisher Scientific, Dreieich, Germany) at a flow rate of 1 mL/min, with the following gradient over 21 min: initial 0.1% (v/v) HCOOH-MeCN/0.1% (v/v) HCOOH-$H_2O$ 0/100, which was increased to 80/20 in 15 min and then to 100/0 in 2 min, held at 100/0 for 2 min, and reversed to 0/100 in 2 min.

## Compound isolation and structure elucidation by NMR

Compounds were isolated by preparative HPLC using a Shimadzu LC-8a series HPLC system with DAD and a Synergi-Fusion RP18 column (Phenomenex, Aschaffenburg, Germany) at a flow rate of 10 mL/min and a gradient system of 0.01% (v/v) TFA ($H_2O$) and MeCN starting at 10% (v/v) MeCN leading to 90% in 20 min. HPLC-HRESI-MS measurements were performed using a Q Exactive hybrid

quadrupole-orbitrap mass spectrometer with an electrospray ion source (capillary voltage 3.2 kV, capillary temperature 250℃) coupled to an Accela HPLC system (Thermo Fisher Scientific, Bremen, Germany). HPLC conditions: C18 column (Accucore C18 2.6 μm 100 mm x 2.1 mm) and gradient elution (acetonitrile/H$_2$O (each containing 0.1% (v/v) formic acid) 5/95 going up to 98/2 in 10 min, then 98/2 for another 4 min; flow rate 0.2 mL/ min). NMR spectra were recorded on a Bruker Avance DRX 600 instrument. Spectra were normalized to the residual solvent signals.

## Analysis of heterologously expressed *A. fumigatus fgn* cluster proteins

Protein preparation and LC-MS/MS analysis and database search for identification of proteins were essentially performed as previously described (*Baldin et al., 2015*), except the following changes: LC gradient elution was as follows: 0 min at 4% B, 5 min at 5% B, 30 min at 8% B, 60 min at 12% B, 100 min at 20% B, 120 min at 25% B, 140 min at 35% B, 150 min at 45% B, 160 min at 60 %B, 170–175 min at 96% B, 175.1–200 min at 4% B. Mass spectrometry analysis was performed on a QExactive HF instrument (Thermo Scientific, Dreieich, Germany) at a resolution of 120,000 FWHM for MS1 scans and 15,000 FWHM for MS2 scans. Tandem mass spectra were searched against the FungiDB database (2020/01/07 https://fungidb.org/common/downloads/Current_Release/AnidulansFGSCA4/fasta/data/FungiDB-46_AnidulansFGSCA4_AnnotatedProteins.fasta) of *Aspergillus nidulans* (FGSC A4) and the protein sequences Afu1g01010, Afu1g01000, Afu1g00990, Afu1g00980, and Afu1g00970 of *Aspergillus fumigatus* Af293 using Proteome Discoverer (PD) 2.2 (Thermo Scientific, Dreieich, Germany) and the algorithms of Sequest HT (version of PD2.2), Mascot 2.4, and MS Amanda 2.0. Modifications were defined as dynamic Met oxidation and protein N-term acetylation as well as static Cys carbamidomethylation.

## Preparation of chromosomal DNA and RNA purification

*A. fumigatus* and *A. nidulans* genomic DNA was isolated as previously described (*Schroeckh et al., 2009*). *A. fumigatus* total RNA was purified with the Universal RNA Purification Kit (Roboklon, Berlin, Germany) according to the manufacturer's instructions.

## Quantitative RT-PCR (qRT-PCR)

Reverse transcription of 5 μg RNA was performed with RevertAid Reverse Transcriptase (Thermo Fisher Scientific, Darmstadt, Germany) for 3 h at 46℃. qRT-PCR was performed as described before (*Schroeckh et al., 2009*). The *A. fumigatus* β-actin gene (*Afu6g07470*) served as an internal standard for calculation of expression levels as previously described (*Schroeckh et al., 2009*). Primers used for the amplification of probes are listed in *Supplementary file 3*.

## Generation of an *A. fumigatus fgnA* deletion strain

The transformation cassettes for the *A. fumigatus fgnA* deletion strain were constructed essentially as previously described (*Szewczyk et al., 2006*). 2000 bp sequences homologous to the regions upstream and downstream of *fgnA* (*Afu1g01010*) were amplified using primer pairs P1/P2 and P3/P4. The pyrithiamine (*ptrA*) resistance cassette was amplified from plasmid pSK275 (*Szewczyk and Krappmann, 2010*) with the primer pair P5/P6. The above PCR-amplified DNA fragments were assembled together with the *Hin*dIII-linearized pUC18 vector using the NEBuilder HiFi DNA Assembly Master Mix (New England Biolabs, Frankfurt, Germany) according to the manufacturer's instructions, yielding plasmid pUC18-*ptrA::fgnA*. The final *fgnA* deletion construct was generated by PCR amplification of the 2 kb upstream and downstream coding regions of *fgnA* flanking the pyrithiamine resistance cassette from plasmid pUC18-*ptrA::fgnA* using primers P1/P4. *A. fumigatus* strain ATCC 46645 was transformed with the resulting linear DNA product as previously described (*Unkles et al., 2014*). Colonies of transformant strains were selected on AMM agar plates supplemented with 0.1 μg/mL pyrithiamine (Sigma-Aldrich, Hamburg, Germany).

## Generation of an inducible *A. fumigatus fgnA* overexpressing strain

For overexpression of *fgnA*, the tetracycline-controlled transcriptional activation system (*tet*$^{On}$) was used (*Helmschrott et al., 2013*) and integrated *in locus* directly upstream of the *fgnA* gene. For this purpose, 2000-bp flanking regions were amplified from *A. fumigatus* ATCC 46645 genomic DNA with the primer pairs P7/P8 and P9/P10, while the *tet*$^{On}$-system was amplified from plasmid pSK562

(Sven Krappmann, personal communication) with primers P11/P12 and the *ptrA* cassette was amplified from plasmid pSK275 with the primer pair P5/P6. The above generated PCR fragments were assembled together with the *Hind*III-linearized pUC18 vector using the NEBuilder HiFi DNA Assembly Master Mix (New England Biolabs, Frankfurt, Germany) according to the manufacturer's instructions, yielding plasmid pUC18-*tet*$^{On}$-*fgnA*. The final transformation construct was generated by PCR amplification from plasmid pUC18-*tet*$^{On}$-*fgnA* using primers P7/P10. The obtained linear DNA fragment, comprising the 2 kb coding regions upstream and downstream of the *fgnA* (*Afu1g1g01010*) ATG start codon, flanking the *tet*$^{On}$ promoter and the *ptrA* cassette, was used for transformation of *A. fumigatus* ATCC 46645. Transformants were selected on AMM agar plates supplemented with 0.1 µg/mL pyrithiamine (Sigma-Aldrich, Hamburg, Germany).

## Heterologous expression of genes of the *fgn* cluster in *A. nidulans*

For overexpression of *fgnA* in *A. nidulans*, the *fgnA* gene was PCR-amplified with primers P13/P14 from either *A. fumigatus* Af293 or ATCC 46645 and fused with the constitutively active *gpdA* promoter of *A. nidulans* amplified together with the pUC18 backbone from plasmid pUC18-*tet*$^{On}$-*basR* (*Fischer et al., 2018*) with primer pair P15/P16. The *A. nidulans argB2* gene was used as a selectable marker to complement the arginine auxotrophy of the *A. nidulans* RMS011 wild-type strain and was amplified together with the *trpC* terminator (T*trpC*) of *A. nidulans* from plasmid pLUC-T*argB*02 (Volker Schroeckh, personal communication) using primer pair P17/P18. All obtained DNA fragments were assembled using the NEBuilder HiFi DNA Assembly Master Mix (New England Biolabs, Frankfurt, Germany) according to the manufacturer's instructions. Transformation of *A. nidulans* was carried out as described before (*Ballance and Turner, 1985*) and transformants were selected on AMM agar plates supplemented with 3 mg/L para-aminobenzoic acid (Sigma-Aldrich, Hamburg, Germany).

For the transfer of the *fgn* cluster from *A. fumigatus* ATCC 46645 to *A. nidulans* RMS011, genes *fgnA*, *Afu1g01000*, *Afu1g00990*, *Afu1g00980*, *Afu1g00970* were PCR-amplified using the primer pairs P19/P20, P21/P22, P23/P24, P25/P26, P27/P28, respectively. Each of the obtained DNA fragments were cloned into a single plasmid. The DNA sequences were separated by 2A sequences, as previously described (*Hoefgen et al., 2018*). The generated plasmid was used for transformation of *A. nidulans* RMS011. Transformants were selected on AMM agar plates supplemented with 3 mg/L para-aminobenzoic acid (Sigma-Aldrich, Hamburg, Germany). As a result, all genes were transcribed as a polycistronic mRNA in *A. nidulans*.

## Southern blot analysis

Southern blotting was performed using a digoxigenin-11-dUTP-labeled (Jena Bioscience, Jena, Germany) probe (*Schroeckh et al., 2009*). The primers used for probe amplification are listed in *Supplementary file 4*.

## Brightfield and fluorescence microscopy

For image acquisition of *Streptomyces* spores or mycelia growing in 48-well plates prepared as described above, a Keyence BZ-X800 microscope was used with 4× magnification for mycelia and 20× phase contrast magnification for imaging of spores. For fluorescence microscopy of fungal mycelia, a Zeiss Axio162 Imager M2-fluorescence microscope (Zeiss, Jena, Germany) was used. *A. nidulans* strains were grown on thin AMM agar plates for 16 h at 37°C, with necessary supplements (*e.g. para*-aminobenzoic acid and doxycycline for the *tet*$^{On}$ inducible mutant strains, or *para*-aminobenzoic acid and arginine for the wild-type strain). An agar plug of 1 cm diameter was removed and placed on a microscopy slide. Nuclei were stained with Hoechst 34580 (Thermo Fisher Scientific, Dreieich, Germany) and visualized with 20× magnification.

## Germination assay

*Streptomyces* spores ($1 \times 10^7$) were incubated in 48-well plates (Thermo Scientific Nunc, Frankfurt, Germany) in presence or absence of fumigermin or germicidin B at 28°C and shaking at 500 rpm in a Plate Shaker (PST-60HL, BioSan, Riga, Latvia). To analyze the impact of the compound on the germination, the $OD_{595}$ of the spore suspension was measured every 24 h for 90 h in a Tecan fluorometer (Infinite M200 PRO, Männedorf, Switzerland). Germicidin B (Cayman Chemical, Ann Arbor, USA) and

methanol were used as controls. The inhibition rate was calculated based on an $OD_{595}$ measurement as previously described (*Petersen et al., 1993*).

## Acknowledgements

We thank Christiane Weigel for antimicrobial assays, Carmen Schult, Silke Steinbach and Karin Burmeister for excellent technical assistance, Heike Heinecke for NMR measurements, and Sven Krappmann and Volker Schroeckh for their kind plasmid donations. Financial support by the International Leibniz Research School for Microbial and Biomolecular Interactions (ILRS), the Deutsche Forschungsgemeinschaft (DFG)-funded Collaborative Research Center 1127 ChemBioSys (projects B01, B02) and the Bundesministerium für Bildung und Forschung (BMBF)-funded project DrugBioTune in the frame of InfectControl2020 is gratefully acknowledged.

## Additional information

### Competing interests

Axel A Brakhage: Reviewing editor, *eLife*. The other authors declare that no competing interests exist.

### Funding

| Funder | Grant reference number | Author |
|---|---|---|
| Deutsche Forschungsgemeinschaft | Collaborative Research Center 1127 ChemBioSys (projects B01) | Christian Hertweck |
| Deutsche Forschungsgemeinschaft | Collaborative Research Center 1127 ChemBioSys (projects B02) | Tina Netzker<br>Axel A Brakhage |
| Bundesministerium für Bildung und Forschung | DrugBioTune | Maria Cristina Stroe<br>Axel A Brakhage |
| International Leibniz Research School | Microbial and Biomolecular Interactions | Axel A Brakhage<br>Maria Cristina Stroe |

The funders had no role in study design, data collection and interpretation, or the decision to submit the work for publication.

### Author contributions

Maria Cristina Stroe, Conceptualization, Data curation, Formal analysis, Validation, Investigation, Visualization, Methodology, Writing - original draft, Writing - review and editing; Tina Netzker, Conceptualization, Validation, Investigation, Writing - review and editing; Kirstin Scherlach, Formal analysis, Investigation, Methodology; Thomas Krüger, Software, Formal analysis, Methodology; Christian Hertweck, Resources, Supervision, Funding acquisition, Writing - review and editing; Vito Valiante, Conceptualization, Resources, Supervision, Methodology, Writing - review and editing; Axel A Brakhage, Conceptualization, Supervision, Funding acquisition, Investigation, Writing - original draft, Project administration, Writing - review and editing

### Author ORCIDs

Maria Cristina Stroe (iD) https://orcid.org/0000-0003-4049-1062
Tina Netzker (iD) http://orcid.org/0000-0003-4845-9366
Vito Valiante (iD) http://orcid.org/0000-0002-4405-169X
Axel A Brakhage (iD) https://orcid.org/0000-0002-8814-4193

### Decision letter and Author response

Decision letter https://doi.org/10.7554/eLife.52541.sa1
Author response https://doi.org/10.7554/eLife.52541.sa2

# Additional files

## Supplementary files

• Supplementary file 1. Putative function of the genes in the predicted PKS biosynthetic gene cluster. Domain analysis using the CD-search tool (NCBI, *Marchler-Bauer et al., 2017*) of each of the cluster genes (shown in black and green), as well as the 5' and 3' neighboring genes. Based on similarity to known genes, the potential function of the encoded proteins was predicted. Gene sequences were downloaded from the FungiDB database (https://fungidb.org/fungidb/).

• Supplementary file 2. Primers used in this study.

• Supplementary file 3. qRT-PCR primers used in this study.

• Supplementary file 4. Primers used for amplification of Southern blot probes.

• Supplementary file 5. Putative orthologs of Fgn cluster proteins identified using BLASTp with a 50% coverage and 50% identity cut-off.

• Transparent reporting form

## Data availability

All data generated or analysed during this study are included in the manuscript. Relevant additional information has been included in the supplementary material.

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
