## [Decision Letter]

**Acceptance summary:**

The reviewers all felt that this study made a significant contribution not only to the biological activity and biosynthesis of an interesting natural product, it also placed the findings in an appropriate evolutionary and ecological context.

**Decision letter after peer review:**

Thank you for submitting your article "Targeted induction of a silent fungal gene cluster encoding the bacteria-specific germination inhibitor fumigermin" for consideration by *eLife*. Your article has been reviewed by three peer reviewers, including Jon Clardy as the Reviewing Editor and Reviewer #1, and the evaluation has been overseen by Gisela Storz as the Senior Editor. The following individuals involved in review of your submission have agreed to reveal their identity: Yi Tang (Reviewer #2).

The reviewers have discussed the reviews with one another and the Reviewing Editor has drafted this decision to help you prepare a revised submission.

Summary:

Stroe et al., utilize a previously established co-culturing method of the fungi *Aspergillus fumigatus* with the bacteria *Streptomyces rapamycinicus* to activate silent gene clusters of *A. fumigatus*. This specific fungi/bacteria pairing has already been previously used to discover product. Nevertheless, the authors were successful in identifying a new methyl substituted, α-pyrone polyketide product, fumigermin, and identify its role as a specific germination inhibitor of *S. rapamycinicus*. Stroe et al., found homologous clusters to the fgn cluster with missense mutations and were able to "repair" these mutations and successfully express fumigermin from these homologous clusters.

Essential revisions:

The main issue with the biosynthetic formulation is the proposal that fumigermin production requires only fgnA – something that would be counter to the reported transcription data and conventional PKS pathway logic. Such a claim requires a higher level of proof than the authors provide. Two referees offer alternative hypotheses that would explain the reported observations in a manner more consistent with accepted PKS biosynthetic pathway logic. They also give suggestions for additional experiments that could clarify the situation – MS networking analysis, additional transcriptional analysis, and/or mutational analysis were all possibilities.

One way or another, the question of whether fgnA acts alone or in a genomic or heterologous host context with other enzymes needs to be clarified.

The nature of the 'germination inhibition' should be described more fully: inhibition or delay, diffusible signal or cell contact.

The authors establish that "a novel fungal compound is triggered by a bacterium," but they do not establish the triggering mechanism. Does cell-free axenic bacterial culture medium induce fumigermin or is the bacteria itself needed? It's a simple experiment and a crucial point. While not necessary, identification of the inducing agent would make a spectacular study.

The speculation about convergent evolution leading to the production of similar compounds by microbes in different species seems likely, but it should be phrased more as a likely possibility, not an established fact. The singularity of a functional cluster does under cut the ecological relevance of the finding. If it really performs an important function, why is the functional version of it so rare? Its rarity argues that the selection benefit is small.

Subsection “The partially reducing FgnA synthesizes fumigermin *via* a PKS I-derived pathway”: Methyl transferase domains typically are referred to as MT domains.

Figure 1A: While the TIC is nice in showing the overall metabolic profile. An EIC of the specific metabolite mass would be more convincing in showing that it does not show up in the individual strains of fungi and bacteria.

Figure 1D: PDA traces should indicate what wavelength is being looked at.

It seems overall for the LC-MS traces, the authors were very inconsistent in showing fumigermin signals, jumping from PDA to TIC to EIC in different figures.

Figure 2A is not sufficient to establish the fgnA-only model. What is needed is a mass spectral network of compounds produced by *A. fumigatus*, and mass spectral networking to find out if molecules with a shared chemical skeleton are synthesized.

Heterologous expression has convincingly shown that the fgn gene cluster is solely required for fungigermin production. Is the biosynthesis in the heterologous host also induced by co-culturing with *Streptomyces*?

Consider adding some more information to the Introduction on work by others on Actinobacteria-Aspergillus interactions. See e.g. Jomori et al., 2019, induction of metabolites by Mycobacterium; Yu et al., 2016, induction of antibacterial cytochalans by cocultivation with *Streptomyces*; Wu et al., 2015, induction of a new polyketide by *Streptomyces* – Aspergillus interactions; Verheecke et al., 2015, on aflotoxin production during cocultivation with *Streptomyces*.

---

## [Author Response]

Essential revisions:The main issue with the biosynthetic formulation is the proposal that fumigermin production requires only fgnA – something that would be counter to the reported transcription data and conventional PKS pathway logic. Such a claim requires a higher level of proof than the authors provide. Two referees offer alternative hypotheses that would explain the reported observations in a manner more consistent with accepted PKS biosynthetic pathway logic. They also give suggestions for additional experiments that could clarify the situation – MS networking analysis, additional transcriptional analysis, and/or mutational analysis were all possibilities.One way or another, the question of whether fgnA acts alone or in a genomic or heterologous host context with other enzymes needs to be clarified.

(Additional experiments 1 and 2): The reviewers raise a valid point, *i.e.* that fumigermin being produced solely by *fgnA* is quite unusual and not according to conventional PKS biosynthetic logic. It is even more surprising since we observed strong transcription of all cluster genes during co-cultivation. In our manuscript we propose that one (or more) of the additional cluster genes (other than *fgnA*) encode non-functional enzymes under the observed conditions.

a) To clarify our arguments, we have explained in more detail the background of the heterologous expression method used (subsection “The *fgnA* polyketide synthase gene is essential for production of fumigermin”). The presence of fluorescence in the nuclei of the transformed *A. nidulans* strain indicated that all 5 proteins of the cluster were produced in the heterologous strain *A. nidulans fgn*_cluster. The wild-type recipient *A. nidulans* strain does not encode any of these *A. fumigatus* genes.

Furthermore, in additional experiment 1 we analyzed the metabolome of the *A. nidulans* transformant strain. Only fumigermin was detected – no additional new metabolite was formed. We have produced a new Supplementary figure depicting the expression strategy and the primary data validating the transformant strain by Southern blot and Northern Blot analysis, as well as the supporting fluorescence microscopy images (Figure 2—figure supplement 1A and B).

b) In additional experiment 2, using label-free LC-MS analysis, we also analyzed the global cytosolic proteome of the *A. nidulans* strain harboring the *A. fumigatus fgn* cluster. The FgnA protein and three putative tailoring enzymes encoded by the *fgn* gene cluster were found in rather high amounts in the *A. nidulans* transformant strain. Importantly, the protein sequences matched the predictions. The fifth cluster gene, *Afu1g00970*, encodes a membrane protein, which, due to its membrane localisation and few trypsin sites, could not be identified in the proteomic analysis. Protein Afu1g00970 could also not be detected in a positive control sample of *A. fumigatus* in co-culture with *S. rapamycinicus*, despite high levels of transcript of the corresponding gene *Afu1g00970*. Nevertheless, the presence of the fluorescence signal in the nuclei of the transformant strain (see a) above) indicated that this protein was also produced along with the others. These additional supporting primary data are found as supplementary information – Figure 2—figure supplement 1 and in the accompanying source data (Figure 2—figure supplement 1—source data 1).

Furthermore, since in this manuscript we focus on the function of fumigermin, we only offer a possible mechanism regarding its biosynthesis. However, several lines of evidence support that FgnA alone can biosynthesize fumigermin:

In non-producer *A. fumigatus* wild-type isolates which encode non-functional *fgnA* variants, no fumigermin production is seen.

Heterologous expression of *fgnA* alone in *A. nidulans* leads to fumigermin production.

* While there might be a slight chance that promiscuous *A. fumigatus* enzymes could act in tandem with FgnA, the probability that similarly promiscuous enzymes would also be found in *A. nidulans* is very low.

Deleting the tailoring enzymes in the native host *A. fumigatus* did not abolish fumigermin biosynthesis.

The fumigermin biosynthesis can be rationalized as being carried out by FgnA only, based on the FgnA enzyme domains, which is further supported by the labelling studies performed.

Previous reports showed that structurally similar compounds like gibepyrone and nectriapyrone are also synthesised by stand-alone PKS enzymes which do not require additional enzymes (Janevska et al., 2016, Motoyama et al., 2019) (see subsection “The fumigermin synthase gene is spread among distantly related fungal classes”).

Taking the reviewers’ comments into account, we have changed our claims regarding FgnA to better reflect the fact that it plays an essential role in the biosynthesis, rather than it being the sole requirement for the biosynthesis of fumigermin (see Abstract, subsection “The *fgnA* polyketide synthase gene is essential for production of fumigermin”).

The nature of the 'germination inhibition' should be described more fully: inhibition or delay, diffusible signal or cell contact.

Thank you for your comment. Fumigermin acts as a reversible inhibitor, *i.e*., the bacterial germination is completely blocked as long as fumigermin is present. When fumigermin was removed, the bacteria once again germinated. We have stated this information more explicitly in subsection “Fumigermin reversibly inhibits the germination of spores of the inducing *S. rapamycinicus.*, Discussion section.

The authors establish that "a novel fungal compound is triggered by a bacterium," but they do not establish the triggering mechanism. Does cell-free axenic bacterial culture medium induce fumigermin or is the bacteria itself needed? It's a simple experiment and a crucial point. While not necessary, identification of the inducing agent would make a spectacular study.

Thank you very much for this hint. We carried out the experiment (see above Experiment 3) and did not find activation of the *fgn* cluster just by supernatant, which is in accordance with our previous findings that physical contact between the two microorganisms is needed. We added a sentence to the manuscript describing these findings (Results section) and also added the additional primary data to the supplement (Figure 1—figure supplement 4).

Heterologous expression has convincingly shown that the fgn gene cluster is solely required for fungigermin production. Is the biosynthesis in the heterologous host also induced by co-culturing with Streptomyces?

(Additional experiment 4): While this is an interesting question, it cannot be analyzed using our experimental system, due to the fact that both heterologous expression strains rely on an artificial expression system of the *fgnA* gene, and that of all *fgn* cluster genes, respectively. The *A. nidulans fgnA* strain harbors the *fgnA* gene under the control of the constitutive *gpdA* promoter, meaning that the *fgnA* gene is constitutively expressed, regardless of the *Streptomyces* partner being present or not. The *A. nidulans fgn* cluster strain harbors all cluster genes under the control of the *tet^On^* promoter. Therefore, addition of doxycycline to the medium activates the gene expression, irrespective of the presence or absence of the streptomycete partner. If doxycycline were not added to the medium, but the streptomycete, there should be no expression of the genes since the genes are not driven by their native *A. fumigatus* promoters. These promoters have been replaced by the *tet^On^* promoter. Nevertheless, to verify this question, we have cultured the *A. nidulans fgn* cluster strain together with the inducer *S. rapamycinicus* and without doxycycline. As predicted, the *fgn* cluster was not activated by the streptomycete in this heterologous host context and the compound was not detected by LC-MS. We have added these data as a supplementary figure (Figure 2—figure supplement 2).

The speculation about convergent evolution leading to the production of similar compounds by microbes in different species seems likely, but it should be phrased more as a likely possibility, not an established fact. The singularity of a functional cluster does under cut the ecological relevance of the finding. If it really performs an important function, why is the functional version of it so rare? Its rarity argues that the selection benefit is small.

Thanks a lot for these very helpful thoughts. We have changed our phrasing accordingly to reflect the fact that convergent evolution is one possible explanation, but certainly not an established fact (subsection “The biosynthesis routes of α-pyrone-based germination inhibitors possibly developed by convergent evolution”). At this stage, we feel it remains rather speculative why the functional version of the gene is so rare. It is conceivable that additional fungal strains or species harboring a functional gene do exist, but have not yet been analyzed in the laboratory and/or genome-sequenced. It is also conceivable that yet unknown compounds produced by other fungi have a similar inhibitory function on the germination of streptomycetes.

Subsection “The partially reducing FgnA synthesizes fumigermin via a PKS I-derived pathway”: Methyl transferase domains typically are referred to as MT domains.

We are very grateful that the reviewers pointed out this issue. We have changed lines 170, 174, Figure 2D and Figure 2—figure supplement 4 to read MT domain.

Figure 1A: While the TIC is nice in showing the overall metabolic profile. An EIC of the specific metabolite mass would be more convincing in showing that it does not show up in the individual strains of fungi and bacteria.

Thank you for this comment. According to your suggestion we have exchanged the TICs for EICs of the fumigermin mass (Figure 1A).

Figure 1D: PDA traces should indicate what wavelength is being looked at.It seems overall for the LC-MS traces, the authors were very inconsistent in showing fumigermin signals, jumping from PDA to TIC to EIC in different figures.

For more consistency in the display of data, and taking into account the previous comment, we have adjusted our representation by changing the PDA traces in Figure 1D to EICs.

Figure 2A is not sufficient to establish the fgnA-only model. What is needed is a mass spectral network of compounds produced by A fumigatus, and mass spectral networking to find out if molecules with a shared chemical skeleton are synthesized.Consider adding some more information to the Introduction on work by others on Actinobacteria-Aspergillus interactions. See e.g. Jomori et al., 2019, induction of metabolites by Mycobacterium; Yu et al., 2016, induction of antibacterial cytochalans by cocultivation with Streptomyces; Wu et al., 2015, induction of a new polyketide by Streptomyces – Aspergillus interactions; Verheecke et al., 2015, on aflotoxin production during cocultivation with Streptomyces.

Thank you for your constructive comment. We have included all the mentioned references and integrated the information of the publications in the Introduction.